# Integrated In Vitro and In Silico Evaluation of the Antimicrobial and Cytotoxic Potential of *Calotropis procera* Leaf Ethanolic Extract: From GC-MS Profiling to Molecular Docking and Dynamics

**DOI:** 10.3390/ijms262110574

**Published:** 2025-10-30

**Authors:** Juan David Rodríguez-Macías, Oscar Saurith-Coronell, Laura Martínez Parra, Domingo César Carrascal-Hernández, Fabio Fuentes-Gandara, Daniel Insuasty, Edgar A. Márquez-Brazón

**Affiliations:** 1Facultad de Ciencias de la Salud, Exactas y Naturales, Universidad Libre Seccional Barranquilla, Barranquilla 080001, Colombia; 2Facultad de Ciencias Básicas, Departamento de Química y Biología, Universidad del Norte, Barranquilla 080003, Colombia; osaurith@uninorte.edu.co (O.S.-C.); domingoh@uninorte.edu.co (D.C.C.-H.); insuastyd@uninorte.edu.co (D.I.); ebrazon@uninorte.edu.co (E.A.M.-B.); 3Estudiante de Maestría en Biotecnología, Universidad Libre Seccional Barranquilla, Barranquilla 080001, Colombia; laura.martinez@unilibre.edu.co; 4Department of Natural and Exact Sciences, Universidad de la Costa, Barranquilla 080002, Colombia; ffuentes1@cuc.edu.co

**Keywords:** *C. procera*, antimicrobial, cytotoxicity, extract, ethanolic

## Abstract

*Calotropis procera*, a drought-tolerant shrub widely used in folk medicine, was evaluated for its antimicrobial potential and safety using an integrative in vitro/in silico workflow. Ethanolic leaf extract (EE-CP) displayed a dose-dependent inhibition of *Staphylococcus aureus* ATCC 2913 and *Escherichia coli* ATCC 35218, reaching 93% and 52% of the amoxicillin control, respectively (MIC 207 µg mL^−1^ and 149 µg mL^−1^). GC-MS and LC-HRMS profiling revealed cardenolides (strophanthidin, gitoxigenin) and indole derivatives as major constituents. Pharmacophore mapping highlighted the essential glycosyltransferase MurG as a likely bacterial target; molecular docking showed that strophanthidin and NCGC00384918 bind MurG more strongly than the native substrate UDP-GlcNAc (ΔG ≤ −9.4 kcal mol^−1^), a result corroborated by 100 ns molecular dynamics simulations and MM-PBSA binding energies (−96.4 and −49.3 kcal mol^−1^). EE-CP caused <10% hemolysis up to 1.5 mg mL^−1^ and exhibited LC_50_ values of 302 µg mL^−1^ (human lymphocytes) and 247 µg mL^−1^ (BHK-21 cells), indicating a narrow but exploitable therapeutic window. Collectively, these findings constitute the first report on Colombian *C. procera* demonstrating potent anti-*Staphylococcus* activity, MurG-targeted cardenolides, and acceptable erythrocyte compatibility. This study supports EE-CP as a promising source of lead molecules and antibiotic adjuvants, warranting guided fractionation and in vivo validation to optimize efficacy and mitigate cytotoxicity.

## 1. Introduction

Medicinal plants have long been recognized as valuable sources of antimicrobial compounds, primarily due to their vast and structurally diverse arsenal of secondary metabolites. These include alkaloids, flavonoids, terpenoids, glycosides, and phenolics, many of which have evolved as part of the plant’s natural defense mechanisms against pathogens. Unlike synthetic antibiotics, which often target a single bacterial protein or pathway, phytochemicals can disrupt microbial viability through multiple modes of action, such as membrane disruption, the inhibition of nucleic acid or protein synthesis, and metal ion chelation, thereby reducing the likelihood of resistance development. This polypharmacological behavior not only enhances their antimicrobial efficacy but also makes plant-derived compounds particularly attractive in the fight against multidrug-resistant pathogens [1,2,3].

Beyond their pharmacodynamic versatility, plant-based antimicrobials offer practical advantages in terms of accessibility, sustainability, and ethnopharmacological validation. In many regions, especially where access to conventional healthcare is limited, traditional medicinal plants such as *Calotropis procera* have been used empirically for centuries to manage infections and inflammatory conditions. These uses, rooted in local knowledge systems, provide a meaningful framework for bioprospecting and scientific validation. Moreover, plants with invasive potential, like *C. procera*, present an ecologically responsible resource for drug discovery, as their exploitation may contribute to ecological control while yielding valuable bioactive molecules. This convergence of ecological utility, biochemical richness, and cultural relevance positions medicinal plants as a vital frontier in the development of next-generation antimicrobials [4,5].

Also known as Sodom apple, calotrope, and giant milkweed, *C. procera* is a shrub native to Africa, the Arabian Peninsula, Western Asia, South Asia, and Indochina. It belongs to the Apocynaceae family, typically grows in arid and semi-arid habitats and reaches a height of 2.5–4 m. This plant, characterized by large pale green leaves and a toxic milky sap in its green fruits, has potential economic and medicinal value, making it of interest for its ecological, economic, and antimicrobial properties [1]. Extracts of *C. procera* have been widely used in traditional medicine due to their pharmacologically active compounds found in their flowers, roots, and leaves, as well as in milky latex. These extracts, particularly from the aerial parts of the plant, have been commonly employed for the treatment of various diseases, including fever, tumors, joint pain, ulcers, muscle spasms, and constipation. The antimicrobial and cytotoxic properties of *C. procera* have been extensively studied, with some researchers suggesting potential applications in developing new eco-friendly antibacterial and antiviral drugs [2].

Recently, Ahmad Nejhad et al. (2023) [3] studied the antimicrobial effects of *C. procera* leaf extracts against *E. coli*, observing a zone of inhibition (ZI) of 14.1 mm. The extracts contained moderate amounts of flavonoids (1781.7 μg/g), phenols (174.82 mg/g), and p-coumaric acid [3]. Another study identified 31 compounds from *C. procera* via gas chromatography-mass spectrometry (GC-MS), including phytol (13.32%), α-amyrin (39.36%), lupeol acetate (17.94%), gombasterol A (2.14%), linolenic acid (3.04%), stigmasterol (3.16%), and hexadecanoic acid (5.55%). The ethanolic extracts demonstrated antimicrobial activity against *Klebsiella pneumoniae*, *E. coli*, and Staphylococcus aureus, with zones of inhibition of 18.66 mm, 21.26 mm, and 21.93 mm, respectively [4].

Previous reports also describe the chemometric profile and antimicrobial activity of *C. procera* leaf extracts, which are rich in palmitic acid ester (10.24%), linoleic acid (7.43%), fatty acid ethyl ester (21.36%) and amino acids (8.1%), with a zone of inhibition of 11 mm against *E. coli* [5]. Additionally, ethanolic and aqueous extracts of *C. procera* leaves and latex have been tested against *Bacillus subtilis*, *Candida albicans*, and *Salmonella typhi* [6,7]. In these studies, ethanolic extracts showed higher efficacy, with a zone of inhibition of 21 mm against *B. subtilis*, whereas the aqueous extracts showed no activity [8].

Given the growing interest in plant-derived antimicrobials, *Calotropis procera* has emerged as a promising candidate due to its rich phytochemical profile and traditional medicinal relevance. While several studies have reported its antimicrobial potential, few have integrated a comprehensive approach that combines both experimental validation and computational modeling to elucidate its bioactive mechanisms.

In this context, the present study offers a novel contribution by not only evaluating the antimicrobial and cytotoxic effects of *C. procera* leaf ethanolic extract through in vitro assays, but also by characterizing its chemical constituents via gas chromatography-mass spectrometry (GC-MS) and assessing their molecular interactions with microbial targets using in silico techniques. Specifically, molecular docking and molecular dynamics (MD) simulations have been employed to predict the binding affinity, stability, and potential mechanism of action of the major phytochemicals identified.

This integrative strategy provides a more robust and mechanistic understanding of the antimicrobial potential of *C. procera*, bridging the gap between empirical observations and molecular-level insights. By combining the strengths of both experimental and computational methodologies, this work not only enhances the predictive power of phytochemical screening but also contributes to the rational design of plant-based antimicrobial agents with improved efficacy and safety profiles.

## 2. Results

### 2.1. Solubility Test for the Ethanolic Extract of C. procera Leaves

A solubility test is a crucial step in the evaluation of the properties and potential applications of the ethanolic extract of *C. procera* leaves [6]. The test consists of evaluating the solubility of the extract in different solvents with different polarities, such as water, chloroform, and ethanol [8]. Important information about the potential pharmaceutical, antimicrobial, and antioxidant properties of a wide range of compounds can be obtained from these tests. For example, previous studies have shown that the ethanolic extract of *C. procera* leaves exhibits a significant inhibition of key enzymes related to diabetes mellitus, indicating its potential for pharmaceutical applications [9,10].

Table 1 reports the solubility results of the ethanolic extracts of *C. procera* leaves; solvents with different polarities such as xylol, chloroform, and heptane were used to deduce the presence of secondary metabolites in the extracts. The extract was fully soluble in ethanol (96%) and dimethyl sulfoxide (DMSO) and partially soluble in methanol. These results suggest that the major constituents of the extract are predominantly polar to moderately polar compounds, which aligns with the known phytochemical profile of *C. procera*, including flavonoids, phenolic acids, and triterpenoids [11].

In contrast, the extract was insoluble in distilled water and acetone, and only slightly soluble in ethyl acetate, chloroform, xylol, and heptane. This limited solubility in non-polar and aprotic solvents indicates a relatively low abundance of highly lipophilic compounds such as long-chain hydrocarbons or non-polar sterols. These findings are consistent with previous GC-MS analyses of *C. procera* extracts, which report a dominance of semi-polar compounds like lupeol acetate, α-amyrin, and phytol [12].

The high solubility in ethanol and DMSO is particularly advantageous for pharmaceutical and antimicrobial applications. Ethanol is widely used in herbal extraction due to its ability to dissolve a broad spectrum of bioactive compounds, while DMSO is a preferred solvent in in vitro assays for its excellent solubilizing capacity and cell permeability. These properties support the extract’s suitability for downstream biological testing, including enzyme inhibition, antimicrobial screening, and cytotoxicity assays [13]. Moreover, the poor solubility in water underscores the importance of solvent selection for both extraction and formulation. This characteristic may also influence the bioavailability and delivery strategies for therapeutic applications, suggesting that encapsulation or emulsification techniques may be required for aqueous formulations [14,15].

### 2.2. Phytochemical Test of the Ethanolic Extract of C. procera Leaves

An extract yield of 8.6% was obtained. The results obtained from the preliminary phytochemical screening of the ethanolic extract of *C. procera* leaves revealed the presence of cardiac glycosides, flavonoids, saponins, alkaloids, tannins, and phenols as shown in Table 2.

The qualitative phytochemical screening of the ethanolic extract of *C. procera* leaves revealed the presence of several key secondary metabolites, including alkaloids, flavonoids, phenols, cardiac glycosides, and saponins, while tannins were absent. These findings are consistent with previous reports on the phytochemical richness of *C. procera*, particularly in its aerial parts.

The detection of alkaloids is noteworthy, as these nitrogen-containing compounds are widely recognized for their antimicrobial, analgesic, and cytotoxic properties. Alkaloids from *C. procera* have been previously reported to exhibit significant antibacterial activity, supporting their potential role in the observed bioactivity of the extract [16,17].

Flavonoids and phenolic compounds, both detected in this extract, are well-known for their antioxidant and antimicrobial properties. These polyphenolic compounds can disrupt microbial membranes and inhibit key enzymes, contributing to the plant’s defense mechanisms. Their presence in *C. procera* has been linked to its traditional use in treating infections and inflammation [18,19,20].

The presence of cardiac glycosides, as indicated by the Kiliani test, is particularly interesting. While these compounds are primarily known for their cardiotonic effects, they have also demonstrated antimicrobial and cytotoxic activities in various plant species. In *C. procera*, cardiac glycosides such as calotropin and uscharin have been isolated and shown to possess potent biological activity [21,22,23].

On the other hand, saponins, detected at a high level (4+ foaming), are amphiphilic glycosides known for their ability to form stable foams and disrupt lipid membranes. Their surfactant properties contribute to antimicrobial and antifungal effects, and they may also enhance the bioavailability of other phytochemicals by increasing membrane permeability [24]. Interestingly, the phytochemistry screening showed an absence of tannins. This result may reflect the solvent selectivity, as ethanol tends to favor the extraction of moderately polar compounds over highly astringent polyphenols. This absence does not diminish the extract’s therapeutic potential but rather highlights the importance of solvent choice in targeting specific metabolite classes.

The presence of alkaloids, flavonoids, phenols, cardiac glycosides, and saponins in the ethanolic extract of *C. procera* leaves provides a strong biochemical basis for many of its traditional medicinal uses. These classes of compounds are frequently associated with antimicrobial, anti-inflammatory, analgesic, and cytotoxic activities, properties that align with the ethnobotanical applications of *C. procera* in treating infections, pain, fever, and inflammatory conditions [25].

### 2.3. High-Performance Liquid Chromatography Coupled to Electrospray Ionization and Quadrupole Time-of-Flight Mass Spectrometry (HPLC-ESI-QTOF-MS)

The HPLC-ESI-QTOF-MS analysis of the ethanolic extract of *Calotropis procera* leaves led to the tentative identification of six compounds, including cardenolides, indole derivatives, and other bioactive phytochemicals (Table 3, Figure 1). These findings provide valuable insights into the chemical complexity and potential pharmacological relevance of the extract.

Among the identified compounds, strophanthidin (C_23_H_34_O_6_) was detected at a retention time of 10.62 min with an experimental m/z of 407.2510. Strophanthidin is a well-known cardiotonic steroid belonging to the cardenolide class, which has been previously reported in *C. procera* latex and leaves [26]. Its presence supports the traditional use of the plant in treating cardiovascular and inflammatory conditions and aligns with earlier reports of cytotoxic and antimicrobial activity associated with cardenolides [27,28,29].

The detection of anthranilate (C_7_H_7_NO_2_), indole-3-carboxylic acid (C_9_H_7_NO_2_), and indole-3-acetic acid (C_10_H_9_NO_2_) further highlights the presence of indole-based metabolites, which are known to play roles in plant defense and microbial inhibition. Indole-3-acetic acid is a plant hormone (auxin) that also exhibits antimicrobial and anti-inflammatory properties [30,31,32].

These compounds may contribute to the observed antimicrobial activity of the extract and suggest a broader ecological role for *C. procera* in its native arid environments.

Two additional compounds labeled NCC20203915 (C_19_H_24_O_4_) and Meli5001779351-O1 (C_16_H_21_N_2_O) were tentatively annotated based on their mass spectra. While these compounds are not yet fully characterized in the context of *C. procera*, their detection suggests the presence of novel or less-studied secondary metabolites. The identification of such compounds opens new avenues for pharmacological exploration, particularly in the context of antimicrobial and cytotoxic screening.

According to date reported in the literature, most metabolomic studies on *C. procera* have been conducted in regions such as India, Saudi Arabia, and North Africa. The current findings represent, to the best of our knowledge, the first report of the HPLC-ESI-QTOF-based metabolite profiling of *C. procera* leaves collected in Colombia. This regional novelty is significant, as environmental factors such as soil composition, climate, and altitude can influence the secondary metabolite profile of medicinal plants [33]. These findings not only validate traditional uses but also provide a chemical foundation for future bioactivity-guided fractionation and drug discovery efforts.

### 2.4. Antibiogram Tests on the Ethanolic Extract of C. procera Leaves

#### 2.4.1. Disk Diffusion Test: *Staphylococcus aureus* and *Escherichia coli*

The selection of *Staphylococcus aureus* (a Gram-positive bacterium) and *Escherichia coli* (a Gram-negative bacterium) as test organisms in antimicrobial assays is both strategic and scientifically justified. These two species are among the most clinically significant pathogens worldwide, frequently implicated in community-acquired and nosocomial infections [34,35,36]. For example, *Staphylococcus aureus* is known for its ability to cause a wide range of infections, from superficial skin conditions to life-threatening diseases such as pneumonia, endocarditis, and sepsis. Its increasing resistance to antibiotics, particularly methicillin-resistant S. aureus (MRSA), has made it a high-priority target for new antimicrobial agents [37,38]. *Escherichia coli*, on the other hand, is a leading cause of urinary tract infections, gastrointestinal diseases, and bloodstream infections. Its capacity to acquire resistance genes, including extended-spectrum β-lactamases (ESBLs), has raised global concern regarding treatment efficacy [39,40,41]. Therefore, by including both Gram-positive and Gram-negative strains, this study ensures a broader evaluation of the antimicrobial spectrum of the *C. procera* extract, which is essential for assessing its potential as a therapeutic agent.

Previous studies have reported the antimicrobial activity of *C. procera* extracts against both *S. aureus* and *E. coli*, although the extent of activity varies depending on the extraction method, plant part used, and geographical origin. For instance, ethanolic extracts of *C. procera* leaves have shown zones of inhibition ranging from 14 to 22 mm against these pathogens, suggesting the presence of potent bioactive compounds such as cardenolides, flavonoids, and alkaloids [42,43,44].

However, to date, there are no published reports on the antibacterial activity of *C. procera* extracts collected in Colombia, making this study the first to explore its efficacy against *S. aureus* and *E. coli* in this regional context. This not only adds to the global phytochemical and pharmacological knowledge of the species but also highlights the potential of Colombian *C. procera* as a source of novel antimicrobial agents.

The extract exhibited dose-dependent antibacterial activity against both pathogens (Table 4, Figure 2). In *S. aureus* ATCC 2913, halo diameters increased from 0.7 ± 0.1 cm at 62.5 µg mL^−1^ to 2.6 ± 0.7 cm at 1000 µg mL^−1^, reaching 93% of the inhibition produced by the positive control (2.8 ± 0.6 cm). The linear fit (R^2^ = 0.97; r = 0.99) indicates an approximate increase of 0.21 cm for every 100 µg mL^−1^ of extract.

In *E. coli* ATCC 35218 the response was similar but less pronounced: diameters rose from 0.7 ± 0.1 cm at 62.5 µg mL^−1^ to 1.6 ± 0.7 cm at 1000 µg mL^−1^, representing 52% of the positive control halo (3.1 ± 0.3 cm). The dose-effect correlation was also high (r ≈ 0.98), with an average increment of 0.18 cm per 100 µg mL^−1^.

No inhibition was detected in the negative control in any assay, confirming the reliability of the method. These findings demonstrate that the extract displays high efficacy against *S. aureus* from 500 µg mL^−1^ onward and moderate activity against *E. coli* only at higher concentrations.

#### 2.4.2. Minimum Inhibitory Concentration Test of *S. aureus* and *E. coli*

The extract exhibited a robust, clearly dose-dependent bactericidal effect: at 1000 µg mL^−1^ it reduced the growth of *S. aureus* by 77% and that of *E. coli* by 65%, approaching the reference effect of the positive control (600 mg amoxicillin), which left only 15% and 8% viability, respectively (see Table 5 and Figure 3). The regression analysis revealed a very strong negative correlation between concentration and growth (r ≥ −0.95; *p* < 0.02), with slopes of −0.21 and −0.18 absorbance units for every 100 µg mL^−1^ increase in extract for *S. aureus* and *E. coli*, respectively. These slopes account for the stepwise absorbance decrease from 0.46 ± 0.02 to 0.07 ± 0.02 in *S. aureus* and from 0.38 ± 0.01 to 0.25 ± 0.05 in *E. coli* when the concentration rose from 62.5 to 1000 µg mL^−1^ (Table 5). The estimated IC_50_ values, 207 µg mL^−1^ for *S. aureus* and 149 µg mL^−1^ for *E. coli*, position the Gram-positive strain as more susceptible at high doses, although the Gram-negative strain requires less extract to achieve 50% inhibition. Standard deviations below 10% of the mean for most treatments underscore experimental precision, while the complete lack of inhibition in the negative control (sterile water) rules out vehicle-related artifacts. These findings confirm that the extract attains 70–90% of the antibacterial potency of amoxicillin against *S. aureus*, shows moderate efficacy against *E. coli*, and emerges as a promising candidate for alternative therapies, particularly targeting resistant Gram-positive pathogens (Figure 3).

### 2.5. Evaluation of the Hemolytic Activity of EEHCP

The ethanolic extract of *C. procera* (EE-CP) caused only marginal damage to human erythrocyte membranes. As summarized in Table 6, hemolysis remained below 10% at every concentration tested. The highest value—9.3 ± 0.1% at 1000 µg mL^−1^—still falls under the generally accepted non-hemolytic threshold, whereas all other doses (32–1500 µg mL^−1^) produced ≤ 2.2% lysis, comparable to the negative control (PBS, 2.2 ± 0.3%; *p* > 0.05). In sharp contrast, the positive control (1% SDS) induced complete membrane rupture (100.6 ± 1.2%). These findings confirm that EE-CP is essentially non-cytotoxic toward human erythrocytes over a 32–1500 µg mL^−1^ range, supporting its biocompatibility for further pharmacological development.

### 2.6. Cytotoxicity of the Ethanolic Extract of C. procera Leaves on Lymphocytes

The ethanolic extract of *C. procera* (EE-CP) elicited a clear, concentration-dependent cytotoxic effect on human lymphocytes after 24 h, as quantified by the trypan blue exclusion assay (Table 7). Viability dropped from 95.0 ± 3.8% in the negative control to 63.1 ± 0.7% at 100 µg mL^−1^ and fell below the positive control level (41.9 ± 5.9%) at 400 µg mL^−1^, reaching a minimum of 33.2 ± 4.5% at 500 µg mL^−1^. Statistically significant cytotoxicity emerged at concentrations ≥300 µg mL^−1^ (*p* < 0.05; Figure 4). Linear regression revealed a strong inverse relationship between extract concentration and cell viability (R^2^ = 0.97), and the resulting equation yielded an LC_50_ of 301.9 ± 10.6 µg mL^−1^. These data demonstrate that EE-CP exerts moderate but measurable toxicity on lymphocytes, with an appreciable loss of viability only at the upper end of the concentration range tested.

### 2.7. Cytotoxicity of Ethanolic Extract of C. procera Leaves on Normal Cell Line Baby Hamster Kidney-21 Cells (BHK-21)

The ethanolic extract of *C. procera* (EE-CP) produced a marked, dose-dependent cytotoxic effect on the non-tumor BHK-21 cell line after 24 h (Table 8; Figure 5). Viability plummeted from 100.5 ± 2.0% in the negative control to 16.7 ± 2.3% at 500 µg mL^−1^—an 83% loss comparable to the positive control injury (6.2 ± 1.1%). Even the lowest concentration tested (62.5 µg mL^−1^) reduced viability to 76.7 ± 17.0%, indicating that all doses differed significantly from the untreated cells (*p* < 0.05). A linear regression of viability versus log-concentration revealed an almost perfect inverse relationship (R^2^ = 0.99), underscoring the consistency of the toxic response. Extrapolation from the regression line yielded an LC_50_ of 246.8 ± 22.6 µg mL^−1^, indicating that half-maximal lethality is reached at roughly one quarter of the highest dose. Collectively, these data demonstrate that EE-CP is substantially more cytotoxic to (BHK-21) cells than to lymphocytes and that its safety margin must be carefully considered in future formulations.

### 2.8. Data Collection

One of the main goals of this study was to determine which compounds in the *Calotropis procera* extract might have antibacterial properties. To do this, we first recreated the 2D structures of these compounds using the JSME molecular editor. This allowed us to generate structure data files (.sdf) for each molecule. We also downloaded .sdf files for a few reference compounds UDP-GlcNAc, β-1,4-GALT1-IN-1, Alendronate, and compound 6 from the PubChem database. Next, we evaluated the drug-like properties of these compounds using Lipinski’s Rule of Five and the ADME-Tox protocol, which looks at how a compound is absorbed, distributed, metabolized, and excreted by the body, as well as its potential toxicity using ProTox-3.0 [45,46]. The results of these assessments are shown in Table 9. Most of the compounds were found to fall into acute toxicity categories 4 to 6. According to the Globally Harmonized System (GHS) for classifying chemicals, there are six levels of acute toxicity. These are based on the LD_50_ value, which estimates the dose needed to kill half of a test population. It is a standard way to measure how toxic a substance is. The GHS considers three main ways a substance can enter the body: by swallowing (oral), through the skin (dermal), or by breathing it in (inhalation). Compounds in category 4, for example, are considered harmful if taken in any of these ways. They typically have an oral LD_50_ between 300 and 2000 mg/kg, a dermal LD_50_ between 1000 and 2000 mg/kg, and an inhalation LD_50_ between 2500 and 5000 ppm. Understanding these toxicity levels is crucial in the early stages of drug development [47]. It helps researchers choose compounds that are less likely to cause harm and more likely to succeed in later testing.

The compounds strophanthidin and NCGC00384918 were classified as toxicity class 2, meaning they are considered fatal if swallowed. Because of this, they could be strong candidates for future structural modifications aimed at reducing their toxicity and improving their safety for potential clinical use. This approach has already been successfully applied in various studies, particularly in the development of anticancer agents [48,49,50].

A notable example is the work conducted to reduce the toxicity and improve the bioavailability of Norcantharidin, a demethylated derivative of cantharidin. Norcantharidin has attracted significant research interest due to its broad range of biological activities, including potential treatments for systemic lupus erythematosus, platelet aggregation disorders, renal interstitial fibrosis, and cancer. However, despite its therapeutic promise, Norcantharidin has been associated with high levels of nephrotoxicity, hepatotoxicity, urotoxicity, and cardiotoxicity. To address these issues, researchers have implemented various structural modification strategies to enhance the compound’s solubility and reduce its toxicity. Interestingly, these modifications have not only improved its safety profile but also led to increased antitumor activity. This highlights how structural changes can positively influence a compound’s ADME properties, potentially boosting its overall biological effectiveness [51].

### 2.9. Target Identification

To explore the potential molecular targets of the principal metabolites identified in *C. procera*, we employed PharmMapper, a robust and widely recognized web-based platform for reverse pharmacophore mapping [52]. This tool predicts likely biological targets by comparing the pharmacophoric features of small molecules against a comprehensive database of pharmacophore models derived from experimentally validated protein structures. PharmMapper has been successfully applied across diverse domains, including natural product research, synthetic drug discovery, and drug repurposing, demonstrating its versatility and predictive power in early-stage drug development [53,54].

Therefore, in our study, the 2D structures of the selected compounds were first optimized to ensure accurate pharmacophore alignment. For each compound, a maximum of 300 potential targets were considered. The selection of potential targets followed a two-step strategy that combined computational confidence with biological plausibility. Initially, PharmMapper predictions were filtered using a normalized score greater than 0.85, ensuring that only targets with a strong pharmacophoric similarity to known ligands were considered. Among these high-scoring candidates, MurG glycosyltransferase was prioritized because it catalyzes the final step in lipid II biosynthesis, a critical precursor for peptidoglycan assembly in bacterial cell walls. This enzyme is indispensable for both Gram-positive and Gram-negative bacteria and is widely recognized as a validated antibacterial target [55]. Previous studies have demonstrated that MurG inhibition results in bactericidal effects in *Escherichia coli* and *Staphylococcus aureus* [56]. The antimicrobial profile observed in this study, marked by strong activity against *S. aureus* and the moderate inhibition of *E. coli*, is consistent with this mechanism, as Gram-positive bacteria are generally more susceptible to disruptions in cell wall synthesis. Furthermore, the cardenolides identified in the extract, such as strophanthidin and gitoxigenin, are known to interact with membrane-associated enzymes, and their high docking scores combined with stable molecular dynamics interactions with MurG reinforce their potential role as inhibitors. Regarding cytotoxicity, cardenolides are well-documented inhibitors of Na^+^/K^+^-ATPase in mammalian cells [57], which explains the narrow therapeutic window observed in the assays of *Calotropis procera* metabolites [58].

By integrating PharmMapper into our computational pipeline, we aimed to bridge the gap between phytochemical identification and mechanistic insight, enabling a more rational and hypothesis-driven evaluation of the antimicrobial and cytotoxic potential of *C. procera* metabolites. The predicted targets, which form the basis for subsequent molecular docking and dynamics simulations, are summarized in Table 10.

The pharmacophoric mapping results presented in Table 10 reveal the most probable protein targets for the compounds, with the highest scoring PharmMapper compounds being NCGC00384918 and strophanthidin. These predictions, based on high-confidence pharmacophore alignments (scores > 0.85), offer valuable insights into the potential mechanisms of action of these metabolites. PharmMapper, a widely used reverse pharmacophore mapping tool, enables the identification of likely biological targets by comparing the structural features of small molecules against a comprehensive database of pharmacophore models derived from experimentally validated protein structures [59].

For NCGC00384918, the top-ranked targets include proteins from *Escherichia coli*, *Pseudomonas aeruginosa*, and *Pseudomonas syringae*, many of which are involved in essential bacterial processes. These include transport system kinases (PDB: 2p67), DNA repair enzymes (PDB: 1ghh), and Type III secretion system (T3SS) proteins (PDB: 1o9y)—the latter being key virulence factors in Gram-negative pathogens and attractive targets for anti-virulence strategies [60,61,62,63,64,65,66].

Similarly, strophanthidin, a well-characterized cardenolide, was predicted to interact with several of the same bacterial targets, including glucosyltransferases (PDB: 1nlm) [67,68,69], nitrate reductases [70,71,72], (PDB: 2nya), and lipoamino dehydrogenases [73,74,75] (PDB: 1lv1). These enzymes are involved in cell wall biosynthesis, nitrogen metabolism, and energy production, all of which are critical for bacterial viability. The recurrence of targets such as glucosyltransferases and transport kinases across both compounds suggests a convergent mechanism of action, potentially enhancing the antimicrobial efficacy of the extract through multi-target synergy, a hallmark of polypharmacology [76].

These computational predictions are consistent with previous in vitro studies demonstrating the antibacterial activity of *C. procera* extracts against both *E. coli* and *Staphylococcus aureus* [77]. The ability of its metabolites to potentially inhibit key bacterial enzymes and virulence factors provides a molecular-level explanation for these observations. For instance, the inhibition of T3SS can attenuate bacterial virulence without necessarily inducing cell death, a promising strategy to reduce selective pressure for resistance. Likewise, targeting glucosyltransferases and nitrate reductases can disrupt fundamental metabolic pathways, leading to bacterial growth inhibition.

Importantly, many of the predicted targets are associated with Gram-negative bacteria, which are notoriously difficult to treat due to their impermeable outer membranes and multidrug resistance mechanisms. The ability of *C. procera* metabolites to potentially engage with such targets underscores their therapeutic promise and supports the traditional use of this plant in managing infectious diseases.

However, the focus was placed on identifying targets from Gram-negative bacteria that play essential roles in survival or replication, particularly those previously recognized as critical for bacterial viability when inhibited. One of the most promising targets identified was the essential glycosyltransferase (MurG). This enzyme belongs to the NDP-glycosyltransferase superfamily and plays a key role in the biosynthesis of glycolipids, glycoproteins, and other important metabolites involved in the bacterial life cycle. Specifically, MurG catalyzes the addition of the sugar nucleotide UDP-GlcNAc to lipid I, forming lipid II, a crucial precursor in the synthesis of peptidoglycan, the main component of the bacterial cell wall [78,79,80]. Compounds derived from vancomycin and other antibiotics, such as Murgocil, have shown strong antibacterial activity by targeting peptidoglycan synthesis in bacteria like *Staphylococcus aureus* and *Escherichia coli*. These compounds interfere with the normal function of MurG, making it a highly attractive target for the development of new antibiotics aimed at disrupting cell wall formation in both Gram-positive and Gram-negative bacteria [81,82,83].

### 2.10. Molecular Docking

The compounds NCGC00384918 and strophanthidin, both derived from the *Calotropis procera* extract and predicted to have potential affinity for the MurG enzyme, were subjected to molecular docking simulations. These were analyzed alongside reference compounds β-1,4-GALT1-IN-6, Alendronate, and compound 6 which have already shown inhibitory activity against MurG in both in silico and in vitro studies. Based on the PharmMapper predictions and the biological relevance of MurG, this enzyme was selected as the primary target for further investigation [84].

As previously mentioned, MurG plays a crucial role in bacterial cell wall synthesis. Any disruption to its function can significantly impair the formation of the peptidoglycan layer, ultimately leading to bacterial cell death [85]. The docking results for the Calotropis compounds and the reference molecules are summarized in Table 11. Strophanthidin showed a binding energy of −9.43 kcal/mol, while NCGC00384918 scored −9.12 kcal/mol. Both values indicate stronger interactions than the enzyme’s natural ligand, UDP-GlcNAc, which had a binding affinity of −8.41 kcal/mol. These findings suggest that the two Calotropis compounds may bind more tightly to MurG than its native substrate, highlighting their potential as promising antibacterial agents.

On the other hand, the reference compounds showed a range of interesting binding affinities. The compound β-1,4-GALT1-IN-6 had a binding energy of −9.29 kcal/mol, which is stronger than that of the natural ligand. This result supports previous in vitro findings where this compound effectively inhibited bacterial growth. In contrast, Alendronate and compound 6 showed lower binding energies of −3.99 and −6.30 kcal/mol, respectively. These values suggest weaker interactions with the MurG enzyme compared with the other compounds. However, it is important to note that a lower binding energy does not necessarily rule out the inhibitory potential of these molecules, as other factors may contribute to their biological activity [86].

To better understand how these compounds interact with MurG, 2D interaction diagrams were generated for the complexes with the strongest binding energies. These visualizations, created using Discovery Studio Visualizer 4.5 (BIOVIA, Dassault Systèmes, San Diego, 2025), are shown in Figure 6 and help illustrate the key intermolecular interactions involved.

As shown in Figure 6, strophanthidin forms hydrogen bonds with key residues ILE245, ARG164, THR266, and GLY191, like the interactions observed with the natural ligand UDP-GlcNAc. This highlights the importance of these amino acids in the inhibition process of the MurG enzyme and suggests a possible shared mechanism of action between these compounds. Likewise, NCGC00384918 also shares hydrogen bond interactions with the natural ligand, particularly with residues SER192, GLU269, and GLY191. These findings point to the potential significance of GLU269 and GLY191 in MurG inhibition. When comparing the interaction profiles of the different MurG inhibitors, it becomes clear that both the Calotropis compounds and β-1,4-GALT1-IN-6, which showed the strongest binding energy among the reference compounds, exhibit a notable number of hydrogen bonds and van der Waals interactions. These involve hydrogen donor oxygens and aromatic rings, suggesting that the number, distance, and type of these interactions play a crucial role in the binding affinity and stability of the enzyme-ligand complexes [87].

The similarity in interaction patterns between the Calotropis compounds (strophanthidin and NCGC00384918), the natural ligand, and β-1,4-GALT1-IN-6 suggests a potentially comparable mechanism of action. This structural and energetic stability supports the idea that these natural compounds could act as effective MurG inhibitors.

### 2.11. Molecular Dynamics

To better understand how stable the ligand-protein complexes are under physiological conditions, we ran molecular dynamics simulations in a water and ion environment that mimics the conditions inside the body [88]. We used the Root Mean Square Deviation (RMSD) method, which is widely applied to track how much the structure of a protein changes over time both in its backbone and in all its heavy atoms [89]. A low RMSD value generally indicates that the protein remains stable and does not undergo significant structural shifts during the simulation. After completing the docking protocol, we focused on evaluating the stability of the complexes formed between the MurG enzyme and three ligands: strophanthidin, NCGC00384918, and the natural substrate UDP-GlcNAc. Each complex was simulated for 100 nanoseconds.

Figure 7 shows the RMSD plots for these MurG-ligand complexes. As illustrated, all three systems, UDP-GlcNAc, strophanthidin, and NCGC00384918, reached a stable equilibrium around 50 nanoseconds into the simulation. Before this equilibration phase, the RMSD values hovered around 2 Å, suggesting that the complexes underwent some initial structural rearrangements to reach a more energetically favorable conformation.

Based on the RMSD values analyzed in Figure 7, the natural ligand UDP-GlcNAc showed an average RMSD of around 1 Å after 50 nanoseconds of simulation, indicating that the system reached a stable equilibrium. Both strophanthidin and NCGC00384918 displayed similar behavior to the natural ligand, suggesting that their interactions with MurG are comparably stable. Notably, strophanthidin stood out with an RMSD of approximately 0.6 Å. After the initial stabilization phase, this complex exhibited only minor conformational fluctuations, and from around 60 ns onward, it maintained the lowest RMSD among all simulated compounds, indicating an even greater binding stability than UDP-GlcNAc.

The stabilization phase during molecular dynamics simulations is crucial for understanding the strength and reliability of ligand binding, which in turn reflects the compound’s potential inhibitory effectiveness [89]. Figure 8 illustrates the binding poses of the simulated compounds at their lowest energy states. As shown, there is a clear structural overlap between the Calotropis compounds and the natural ligand, suggesting that they may share a similar binding mode. This overlap supports the idea that these natural compounds could act as effective inhibitors of the MurG enzyme.

Figure 9 shows the 2D interaction diagrams of the complexes at their lowest energy states. As mentioned earlier, the main types of interactions between MurG and its natural ligand are hydrogen bonds and van der Waals forces. What stands out is the preservation of certain key interactions throughout the simulation. For example, both the natural ligand and strophanthidin consistently maintained a hydrogen bond with the amino acid residue GLU269. This suggests that this specific interaction may play a crucial role in stabilizing the complex and could potentially enhance the compound’s biological activity.

To better understand how the ligands interact with the MurG enzyme, we calculated their binding energies using the MM-PBSA (Molecular Mechanics Poisson-Boltzmann Surface Area) approach. This method provides a more detailed view of the interaction process by analyzing the molecular dynamics simulation trajectories [90,91,92]. These simulations were carried out using the YASARA structure 2025.1 which includes a macro called md_analyzebindenergy.mcr. This macro estimates binding energy using Equation 1.

It is important to note that in YASARA’s default setup, more positive values indicate stronger binding affinities. However, negative values do not necessarily mean there is no interaction as they can still reflect meaningful binding, depending on the context. To make the interpretation of our results more intuitive and consistent with previous studies, we inverted and rewrote the original equation from the YASARA macro, as shown in Equation 2 [93].(1)$$Bind\;Energy\;=\;E_{Recept\left(E_{solv}+E_{pot}\right)}+E_{Ligand\left(E_{solv}+E_{pot}\right)}-\;E_{Complex\left(E_{solv}+E_{pot}\right)}$$where E*_Recept_* is the receptor (protein) energy which is the sum of the solvation and potential contributions; E*_ligand_* is the energy of the ligand, and E*_complex_* is the energy of the ligand-receptor (protein) complex.(2)$$Bind\;Energy\;=\;E_{Complex\left(E_{Solv}+E_{Pot}\right)}-\;E_{Ligand\left(E_{Solv}+E_{Pot}\right)}-E_{Recept\left(E_{Solv}+E_{Pot}\right)}$$

This macro calculates the average binding energy by using the solvation energy (E*_Solv_*) and potential energy (E_Pot_) of the ligand, the receptor, and the ligand-receptor complex. The results of these binding energy calculations are summarized in Table 12.

As shown in Table 12, the binding energies between MurG and the different simulated compounds were calculated and compared. UDP-GlcNAc was used as a reference point for interpreting the results, since binding energy is a key indicator of complex stability. Lower values generally suggest more stable interactions, which often correlate with better biological activity.

UDP-GlcNAc showed a binding energy of −237.56 kJ/mol. While this value is positive, as mentioned earlier, it does not mean that there is no stable interaction. Instead, it reflects the estimated stability of the ligand-protein complex. Strophanthidin, on the other hand, had a binding energy of −96.44 kJ/mol, which is significantly lower than that of the natural ligand, suggesting a more stable and potentially stronger interaction over time. Even more promising was NCGC00394918, which achieved the lowest binding energy of −49.34 kJ/mol in the molecular dynamic simulations.

Both Calotropis compounds outperformed the natural ligand in terms of binding energy. NCGC00394918 showed the most favorable result, followed by strophanthidin. These findings suggest that these natural compounds may play a role in inhibiting bacterial growth and could be strong candidates for the development of new antibiotics. To better visualize how binding energy fluctuates over the course of the simulation, Figure 10 presents a graph showing the changes in interaction energy over time.

As shown in Figure 10, UDP-GlcNAc exhibited only minor fluctuations throughout the simulation, indicating a stable interaction with the MurG protein. This aligns with the RMSD plot, where the complex reached a more energetically favorable and stable state around 25 ns and maintained that equilibrium for the remainder of the simulation. A similar pattern was observed with strophanthidin, which stabilized around 30 ns and remained steady, suggesting a comparable level of interaction stability to the natural ligand. This supports the idea that strophanthidin may possess inhibitory properties against MurG.

In contrast, NCGC00384918 displayed a different behavior. Although it showed the lowest average binding energy around −49.34 kJ/mol, its interaction energy dropped sharply around 45 ns, likely due to conformational changes in the protein-ligand complex that temporarily weakened the interaction. However, from 60 ns onward, the binding energy gradually increased, eventually surpassing the initial values. This suggests a structural rearrangement that led to more favorable interactions. This trend is consistent with the RMSD data, which also showed a decrease around the same time, indicating a more stable complex. These findings provide strong evidence that both strophanthidin and NCGC00384918 are promising antibacterial candidates. The results support the hypothesis that their primary mechanism of action involves the inhibition of the MurG glycosyltransferase, an essential enzyme in the peptidoglycan biosynthesis pathway, which is critical for bacterial survival.

While these in silico results align with previous in vitro findings, further validation is necessary. Specific experimental assays are needed to confirm the direct interaction of these compounds with MurG and to fully understand their mechanism of action. It is also important to acknowledge the limitations of computational models. These simulations often simplify biological systems, which can lead to discrepancies between predicted and actual interactions. Additionally, the accuracy of these models depends on the quality of the structural data and the simulation conditions, which may not fully reflect real biological environments [94,95]. Therefore, experimental confirmation is essential. Further molecular modifications should also be explored to reduce toxicity, minimize side effects, and enhance the efficacy of these candidate molecules [96,97]. Overall, the approach used in this study provides valuable insights into the biological systems involved and offers a strong foundation for the development of new antibacterial agents

## 3. Discussion

*Calotropis procera* (“giant milkweed”) has shifted from an ornamental curiosity to a high-risk invasive species in Colombia. Herbarium records and iNaturalist observations already place it in at least ten departments across the Caribbean lowlands, the Andes, and the Magdalena Valley, colonizing roadsides, coastal dunes, and arid savannas thanks to its tolerance of drought and salinity. Vigorous resprouting after cutting or fire, wind-dispersed cotton-like seeds, and allelopathic effects make control challenging [98,99,100].

Yet the plant is still exploited locally: latex for treating warts, leaves as an analgesic, stems for fiber and fuel, and even as an alternative host for monarch butterflies. Colombian research, however, is almost nonexistent, being limited to a single undergraduate thesis with preliminary antimicrobial assays. No peer-reviewed national studies address its phytochemistry, toxicity, or ecological impacts. This gap contrasts sharply with numerous foreign reports highlighting its antibacterial, antifungal, and cytotoxic activities. The findings presented here therefore provide the first local evidence and are essential for weighing ecological risks against its pharmacological promise [98,101,102,103].

The remarkable breadth of secondary metabolites in *Calotropis procera* underpins the robust antibacterial activity observed for its crude extracts and purified fractions. Phytochemical profiling by GC-MS and HPLC consistently reveals high levels of methoxylated flavonoids (e.g., quercetin- and isorhamnetin-glycosides), triterpenes such as α/β-amirine and lupeol, and a rich arsenal of cardenolides including calotropina, calactina, and uzarigenina. Added to this small-molecule repertoire are a suite of bioactive proteins and enzymes—most notably a 14.4 kDa latex lysozyme (LL) and several cysteine proteases (procerain, CpCP1-3)—that together endow the plant with a multi-modal defense system.

This chemical complexity translates into a broad yet differential antibacterial spectrum. Crude ethanolic extracts of the leaves suppress *Staphylococcus aureus* (MRSA) with IC_50_ values close to 200 µg mL^−1^, reaching up to 93% of the inhibition achieved by amoxicillin, while *Escherichia coli* and *Pseudomonas aeruginosa* require higher doses (MIC ≥ 1 mg mL^−1^) to elicit comparable effects. In contrast, the purified latex lysozyme exhibits single-digit MICs (13–14 µg mL^−1^) and inhibition zones of 15–24 mm against a panel of 20 pathogenic strains, including multidrug-resistant *Klebsiella* and *Salmonella* species. Electron-microscopy studies confirm that LL perforates the peptidoglycan meshwork, causing catastrophic membrane collapse and the leakage of cytoplasmic contents, whereas flavonoids and triterpenes appear to compromise membrane integrity through lipophilic insertion, the chelation of essential metals, and reactive oxygen modulation. Unsaturated fatty acid methyl esters present in flower and leaf extracts provide a further layer of membrane destabilization, broadening the antibacterial portfolio of the crude drug.

Importantly, several fractions act as resistance modulators. Ethanolic leaf extract lowers the MIC of gentamicin against *S. aureus* by 65% and of imipenem against *P. aeruginosa* by almost two orders of magnitude, while remaining antagonistic to certain fluoroquinolones. Such synergistic interactions point to the extract’s value as an antibiotic adjuvant capable of resensitizing recalcitrant pathogens. The overall pattern that emerges is one of preferential potency toward Gram-positive bacteria—owing to easier access to the peptidoglycan layer—but with the capacity, through enzymatic or combinatorial mechanisms, to overcome the outer membrane barrier of Gram-negative species.

Taken together, these findings position *C. procera* as a chemically versatile source of antimicrobial leads. The coexistence of potent enzymatic weapons (latex lysozyme), membrane-active flavonoids and triterpenes, and high-affinity cardenolides offers multiple points of attack against resistant pathogens. However, the narrow therapeutic window noted for some crude fractions and the higher cytotoxicity toward mammalian BHK-21 cells underscore the need for guided fractionation and structural optimization. The targeted isolation of the most active yet least cytotoxic constituents, coupled with in vivo validation and mechanistic dissection, will be essential before *C. procera*-derived molecules can progress toward pre-clinical development as standalone antimicrobials or synergy-boosting adjuvants.

Multiple reports concur that the antimicrobial activity of *Calotropis procera* depends on both the extraction solvent and the class of metabolites concentrated. In the literature, ethanolic and methanolic extracts—rich in methoxylated flavonoids, triterpenes, and cardenolides—produce inhibition zones of 8.5–35.4 mm and MIC values ranging from 0.6 µg mL^−1^ (latex lysozyme) to 40 mg mL^−1^ in semipolar fractions, clearly outperforming aqueous extracts, whose potency seldom falls below 6 mg mL^−1^.

The present study confirms this trend: the ethanolic extract generated maximum halos of 2.6 ± 0.7 cm (≈26 mm) against *Staphylococcus aureus* and 1.6 ± 0.7 cm against *Escherichia coli*, equivalent to 93% and 52% of the activity of amoxicillin, respectively. The IC_50_ values obtained (207 µg mL^−1^ for *S. aureus* and 149 µg mL^−1^ for *E. coli*) lie at the lower end of the MIC values reported for crude extracts, confirming that the ethanolic preparation concentrates on the most active compounds.

The comparative target analysis is equally enlightening. Previous docking studies identified binding affinities ≤ −10 kcal mol^−1^ for lupeol, L-rhamnose, and calotroproceryl acetate A toward the *S. aureus* tyrosyl-tRNA synthetase, and values of −6.0 to −6.7 kcal mol^−1^ for root diterpenes against *E. coli* DNA gyrase B. In this work, the cardenolides strophanthidin (−9.43 kcal mol^−1^) and NCGC00384918 (−9.12 kcal mol^−1^) exhibited even stronger affinities for the MurG glycosyltransferase. Moreover, 100 ns simulations revealed stable complexes (RMSD < 2 Å) and far lower MM-PBSA energies (−96.4 and −49.3 kcal mol^−1^) than the natural ligand, with GLU269 acting as a key anchor residue—strengthening MurG’s candidacy as a therapeutic target in both Gram-positive and Gram-negative bacteria.

Regarding safety, hemolysis remained below 10%, between 32 and 1500 µg mL^−1^, and lymphocytes exhibited lower cytotoxicity (LC_50_ ≈ 302 µg mL^−1^) than BHK-21 cells (LC_50_ ≈ 247 µg mL^−1^), confirming the relatively narrow therapeutic window reported for other extracts yet supporting the initial biocompatibility of the ethanolic fraction. Together with the in vitro synergy reported with gentamicin and imipenem, these results endorse cardenolide-enriched fractions as potential adjuvants or lead antimicrobial prototypes against resistant Gram-positive and Gram-negative pathogens, provided that structural optimization is pursued to mitigate the cardiotoxicity typical of this family.

## 4. Materials and Methods

### 4.1. Preparation of the Ethanolic Extract of Calotropis procera

Four kilograms of fresh leaves of flowering *C. procera* leaves were collected during the rainy season on the Universidad Libre campus, Barranquilla (11°01’19” N 74°51’54” W). After discarding damaged material, 3325.5 g were dehydrated in an oven at 60 °C to a constant weight, yielding 331.2 g of dry leaves (≈90% moisture loss) [1]

The dried leaves were ground to a powder and mixed with 500 mL of absolute ethanol (Sigma-Aldrich, St. Louis, MO, USA) under gentle stirring for 24 h. The ethanolic supernatant was filtered, and the solvent was removed on a rotary evaporator (40 °C, 120 mbar), producing a green semi-solid residue. This residue was resuspended in sterile DMSO (Merck, Darmstadt, Germany) to 10 mg mL^−1^ (final DMSO ≤ 1% *v*/*v*) and passed through a 0.22 µm membrane filter.

The resulting ethanolic extract of *C. procera* leaves (EEHCP) was stored at 4 °C and used directly in the biological assays described in this study [104].

### 4.2. Disk Diffusion Assay

Eighteen-hour cultures of both bacteria were adjusted to 0.5 McFarland and lawn-spread on Mueller-Hinton agar (Millipore, Merck, Darmstadt, Germany). Sterile paper disks (6 mm) were impregnated with 10 µL of extract at 1000, 500, 250, 125, and 62.5 µg mL^−1^ and placed on the agar alongside amoxicillin disks (10 µg, positive control) and disks containing 10 µL of 1% DMSO (negative control). Plates were incubated for 24 h at 37 °C and inhibition zones (mm) measured with a digital caliper. All assays were run in independent triplicate [105,106].

### 4.3. Minimum Inhibitory Concentration (MIC)

MIC values were determined by broth microdilution in Mueller-Hinton broth following CLSI M07-A10. The concentrations of the extracts used in our bioassays were prepared starting from a stock solution at 20,000 ppm (μg/mL). An aliquot of this stock was added into the medium to achieve the desired test concentrations. When further dilutions were necessary, additional medium was used to adjust the concentration accordingly. This approach ensured that the extracts were initially dissolved in a solvent and then appropriately diluted into the testing medium for each experiment.

Two-fold serial dilutions of the extract (62.5–1000 µg mL^−1^) were prepared in sterile 96-well plates. Wells were inoculated with 100 µL of *S. aureus* ATCC 2913 or *E. coli* ATCC 35218 suspensions adjusted to 0.5 McFarland and diluted to 1:100, yielding 5 × 10^5^ CFU mL^−1^. After 18 h at 37 °C, the MIC was recorded as the lowest concentration without visible turbidity. The absorbance was measured at 600 nm. Amoxicillin 10 µg mL^−1^ and broth + 0.5% DMSO were used as positive and negative controls, respectively [107]. The absorbance values obtained in the assays were converted to percentages using the negative control as the reference, since it represents the maximum bacterial growth in this medium. Although this does not alter the MIC value, it enables a clearer interpretation of the results.

### 4.4. Cytotoxicity in Human Lymphocytes

Peripheral blood was collected from healthy donors. After a 1:1 dilution with PBS, mononuclear cells were isolated on a Ficoll-Paque™ gradient (400× *g*, 30 min), washed twice and resuspended in RPMI-1640 (Sigma-Aldrich, St. Louis, MO, USA) containing 10% FBS (Sigma-Aldrich, St. Louis, MO, USA) and 1% penicillin/streptomycin. Cell suspensions (1 × 10^6^ cells mL^−1^) were incubated for 24 h with the extract (100–500 µg mL^−1^). Viability was assessed by trypan blue exclusion, mixing 10 µL of cell suspension with 10 µL of 0.4% trypan blue and counting viable/non-viable cells in a Neubauer chamber. RPMI + 0.1% DMSO served as the negative control and SDS 0.05% *w*/*v* as the positive control. LC_50_ was determined by the linear regression of the dose-response curve [108,109].

### 4.5. Cytotoxicity of the Extract on the BHK-21 Cell Line

BHK-21 cells (ATCC^®^ CCL-10™) were maintained in RPMI-1640 supplemented with 10% FBS, 1% penicillin/streptomycin (PAA, Toronto, ON, Canada) (100 U mL^−1^/100 µg mL^−1^), and 2 mM L-glutamine at 37 °C, with 5% CO_2_. Cells (1 × 10^4^ per well) were seeded in 96-well plates; after 24 h adherence they were treated with the extract (62.5–500 µg mL^−1^, 1:2 dilutions). Following 24 h exposure, the medium was removed, 100 µL of MTT solution (Sigma-Aldrich, St. Louis, MO, USA) (0.5 mg mL^−1^ in PBS) was added and the plates were incubated for 4 h. Formazan crystals were dissolved in 100 µL DMSO and absorbance read at 570 nm. The negative control was RPMI + 0.1% DMSO; the positive control was 1 mM H_2_O_2_ [110]. Viability percentages were calculated relative to the negative control and LC_50_ values obtained by logit fitting (IBM^®^ SPSS^®^ Statistics, version 22).

### 4.6. Hemolytic Activity

Human erythrocytes (EDTA anticoagulated) were washed three times with PBS (pH 7.4) and adjusted to a 4% suspension (*v*/*v*). A 100 µL aliquot of this suspension was mixed with 100 µL of extract (32–1500 µg mL^−1^) and incubated for 60 min at 37 °C. After centrifugation (1000× *g*, 5 min) the supernatant absorbance was recorded at 540 nm [111]. Percentage hemolysis was calculated using PBS as the negative control (NC) and 1% SDS as the positive control (PC) as follows:% Hemolysis = (Asame − ACN)/(ACP − ACN) × 100(3)

Samples with < 10% hemolysis were considered biocompatible.

### 4.7. Data Collection

To begin, the molecular structures of the compounds found in the *Calotropis procera* extract were modeled using the JSME platform, a JavaScript-based molecular structure editor that allows for interactive and intuitive molecule design. Using this tool, the structures were exported in .sdf format for further analysis [112,113]. Meanwhile, the structures of UDP-GlcNAc, β-1,4-GALT1-IN-6, Alendronate, and compound 6 were retrieved directly from the PubChem database. The 3D structures of all these compounds including those from *Calotropis procera* and the reference molecules were then generated using OpenBabel 3.0.0. [114]. A genetic algorithm was applied to optimize the molecular conformations, followed by energy minimization using the MMFF94s force field. The final energy-minimized structures were saved in .pdb format and prepared for molecular docking simulations [115].

### 4.8. Target Identification

To identify potential molecular targets, we used the PharmMapper platform (http://lilab.ecust.edu.cn/pharmmapper/; accessed on 2 September 2025). This software begins by generating 3D conformations of the input molecules provided in .sdf format. An MMFF94 force field is first applied to adjust the energy parameters, which are then used to generate up to 300 conformations per molecule using a genetic algorithm. Target prediction is carried out using a druggable pharmacophore model, which identifies likely protein targets for small molecules based on reverse pharmacophore matching. This process involves screening against several databases, including DrugBank, BindingDB, PDBBind, and PDTD, which together contain around 7000 receptor-based pharmacophore models. For each compound, the top 300 predicted targets were retrieved [116].

From these results, we selected only those targets with a normalized score higher than 0.85. The selected targets were then organized into tables, ranked in descending order of binding probability. Special attention was given to targets from Gram-negative bacteria, particularly *Escherichia coli*, prioritizing those involved in essential biological functions.

### 4.9. ADME-Tox Properties

The term ADME-Tox refers to a set of key properties that describe how a compound behaves in the body: absorption, distribution, metabolism, excretion, and toxicity. These characteristics are essential in drug development because they help researchers understand how a drug will act once it’s inside the body [117]. ADME-Tox analysis also helps define the safety profile of a compound, including the safest route of administration and the dosage range within which it can be used without causing harm [118]. By comparing a compound’s ADME-Tox properties to those of known substances in established databases, researchers can quickly identify molecules with poor pharmacokinetic or toxicological profiles. This allows them to either discard unsuitable candidates early on or make structural modifications to improve the safety and effectiveness of promising compounds that have shown good biological activity in silico or in vitro [119].

For this study, we used the ProTox 3.0 web platform to evaluate the ADME-Tox profiles of the compounds found in the *Calotropis procera* extract. The results were interpreted using the classification guidelines from the Globally Harmonized System (GHS) for chemical labeling and safety [120,121,122]. According to these guidelines, special attention was given to compounds classified under toxicity class 2, which are considered potentially fatal at doses between 5 and 50 mg/kg. These were flagged as high risk and treated with caution during the analysis. It also points out which molecules might need to be tweaked to make them safer and helps determine the best way to deliver them in a potential treatment [123].

### 4.10. Docking Protocol

Molecular docking is a powerful computational technique used to predict how two molecules, typically a ligand and a target protein, might interact. It plays a key role in drug discovery by helping identify new therapeutic candidates and potential mechanisms of action. Thanks to its ability to screen large libraries of compounds quickly and efficiently, molecular docking has become one of the most widely used tools in modern drug development. It is also valuable for uncovering unknown therapeutic targets and predicting possible side effects [124,125]. In this study, we used AutoDock-GPU 1.6 (AD-GPU) and AutoDock Vina 1.2.0 to analyze the molecular interactions between the compounds found in the *Calotropis procera* extract, UDP-GlcNAc, and known glycosyltransferase inhibitors. AD-GPU leverages GPU hardware to significantly speed up docking simulations by implementing a gradient-based search algorithm known as ADADELTA [126,127,128].

To better understand the potential mechanism of action of the Calotropis compounds, we first searched for molecules previously reported to inhibit glycosyltransferase activity both in silico and in vitro. From this search, three reference compounds were identified: β-1,4-GALT1-IN-1, Alendronate, and compound 6. These were used as benchmarks to validate the docking results of the *C. procera* compounds [129].

The ligands and target protein were prepared using AutoDockTools and OpenBabel. [130] First, water molecules and any non-essential compounds not part of the primary structure of the protein or ligand were removed. Then, Gasteiger charges were calculated and added to ensure proper electrostatic interactions during the docking simulations. Both ligands and targets were saved in .pdbqt format for compatibility with the docking software. For molecular docking studies, ligand flexibility was enabled. This allowed the simulation to account for possible conformational changes the ligand might undergo to achieve a more energetically favorable interaction with the protein. Meanwhile, the protein was kept rigid throughout the simulation. The docking process was carried out using a custom AutoDock-GPU script, employing the ADADELTA search algorithm. For each docking run, 100 different conformations were generated, based on a maximum of 42,000 generations and 2,500,000 evaluations. The center and dimensions of the docking grid box are detailed below in Table 13.

### 4.11. Molecular Dynamics

To evaluate how well the candidate compounds might perform as potential drugs, it is essential to understand the stability of their interactions with their target proteins. Molecular dynamics (MD) simulations are a widely used computational technique that models how proteins and small molecules behave under physiological conditions [131,132]. These simulations help assess the type, strength, and duration of intermolecular interactions within a ligand-receptor complex. The insights gained from MD simulations are crucial for identifying compounds with the most favorable binding properties [133]. Beyond drug discovery, MD simulations are also valuable for studying how mutations affect protein behavior and their links to various diseases. They are especially useful in understanding how mutations influence drug-receptor interactions, which are critical in contexts like antibiotic resistance [133].

For this study, we used the software YASARA Dynamics 32.12.24 and the AMBER14 force field to run the simulations. Each complex was placed in a cubic simulation box with a 5 Å buffer around the system’s atoms and periodic boundary conditions were applied. To mimic physiological conditions, the system was set at pH 7.4, a temperature of 298 K, and a pressure of 1 atm. A 0.9% NaCl ion concentration was used, and the system was solvated with water at a density of 0.997 mg/mL. Each simulation was run for 100 nanoseconds. After the simulations, we analyzed the results using YASARA’s md_analyze.mcr macro, which calculates key structural parameters such as RMSD (Root Mean Square Deviation) to assess protein stability. We also used the md_analyzebindenergy.mcr macro to estimate binding energy, applying a modified Poisson-Boltzmann equation. In this context, more negative binding energy values indicate stronger, more stable interactions often correlating with a higher potential efficacy of the compound.

These results allow us to compare the performance of different candidate drugs against the natural ligand, helping to identify the most promising molecules for further development.

## 5. Conclusions

The present study provides compelling evidence that the ethanolic extract of *Calotropis procera* leaves possesses significant antibacterial activity, particularly against Staphylococcus aureus, a pathogen of high clinical relevance. Through a combination of in vitro assays and in silico modeling, we demonstrated that the extract not only inhibits bacterial growth in a dose-dependent manner but also interacts favorably with key molecular targets such as MurG glycosyltransferase, an enzyme essential for peptidoglycan biosynthesis. In addition, the integration of high-performance liquid chromatography coupled to electrospray ionization and quadrupole time-of-flight mass spectrometry (HPLC-ESI-QTOF-MS) allowed for the identification of bioactive cardenolides, including strophanthidin and NCGC00384918, which exhibited strong binding affinities and stable interactions with MurG in molecular dynamics simulations. These findings suggest a plausible mechanism of action and reinforce the therapeutic potential of these compounds. From a safety perspective, the extract showed acceptable hemocompatibility, with hemolysis remaining below the 10% threshold across tested concentrations. However, cytotoxicity assays revealed a relatively narrow therapeutic window, particularly in non-tumor BHK-21 cells, highlighting the need for careful dose optimization and further fractionation to isolate the most promising constituents. Finally, this work represents the first comprehensive report on Colombian *C. procera* integrating phytochemical profiling, antimicrobial screening, cytotoxicity evaluation, and computational modeling. The results underscore the plant’s potential as a source of antimicrobial leads and antibiotic adjuvants, especially in the context of rising resistance among Gram-positive pathogens. Future studies should focus on guided fractionation, the structural modification of lead compounds to reduce toxicity, and in vivo validation to advance these findings toward preclinical development.

## Figures and Tables

**Figure 1 ijms-26-10574-f001:**
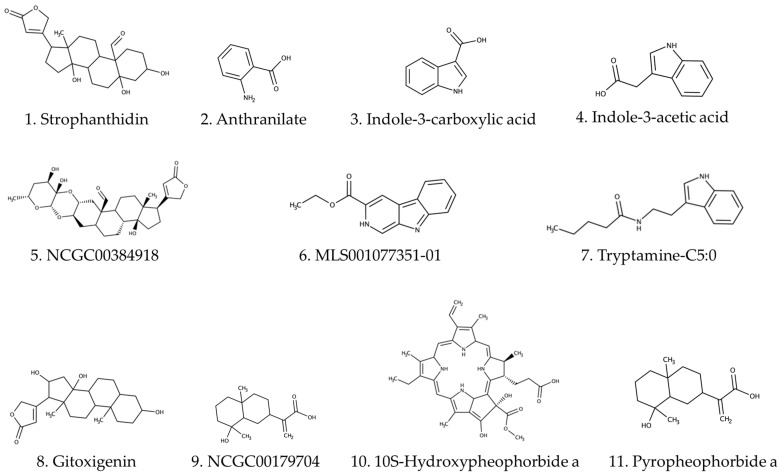
Chemical structures of the eleven compounds tentatively identified by LC-HRMS. The illustration depicts the molecular structures corresponding to the main LC-HRMS signals: (**1**) strophanthidin, (**2**) anthranilate, (**3**) indole-3-carboxylic acid, (**4**) indole-3-acetic acid, (**5**) NCGC00384918, (**6**) MLS001077351-01, (**7**) tryptamine-C5:0, (**8**) gitoxigenin, (**9**) NCGC00179704, (**10**) 10S-hydroxypheophorbide a, (**11**) pyropheophorbide a.

**Figure 2 ijms-26-10574-f002:**
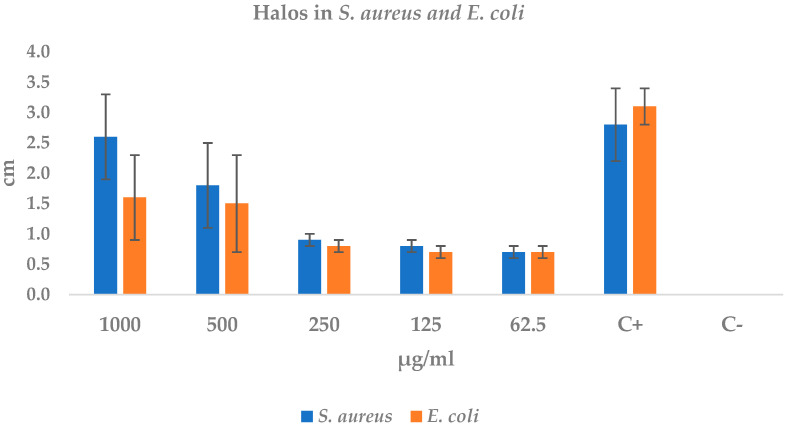
Bar graph of the average diameter of *S. aureus* and *E. coli*. halos compared to positive control (C+: Amoxicillin 600 mg) and negative control (C−: sterile water).

**Figure 3 ijms-26-10574-f003:**
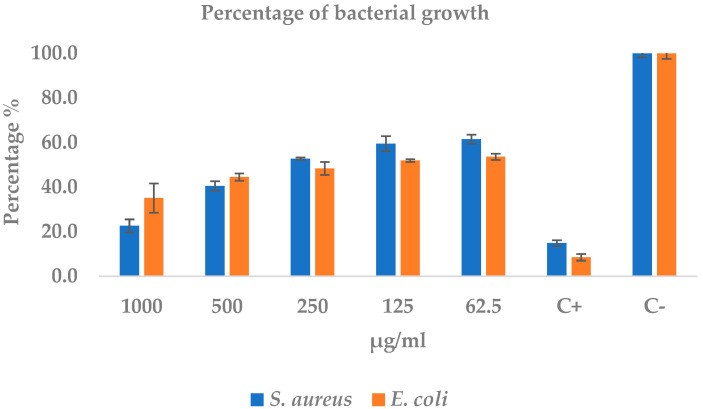
Percentage growth of *S. aureus* and *E. coli* after 24 h exposure to the extract. Blue bars (*S. aureus*) and orange bars (*E. coli*) represent residual growth (mean ± SD) normalized to the negative control (C-; sterile water, 100%). Five extract concentrations (62.5–1000 µg mL^−1^), a positive control (C+; amoxicillin 600 mg L^−1^), and the negative control were evaluated. Error bars denote the standard deviation. Lower values indicate greater inhibitory activity.

**Figure 4 ijms-26-10574-f004:**
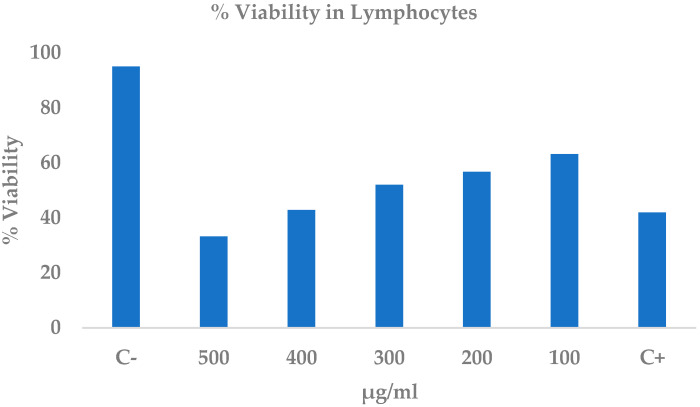
Percentage of cell viability at different concentrations of ethanolic extract of *C. procera* leaves on human lymphocytes at 24 h. C+: positive control (methanol), C−: Negative control (PBS). The results shown are an average of 5 experiments.

**Figure 5 ijms-26-10574-f005:**
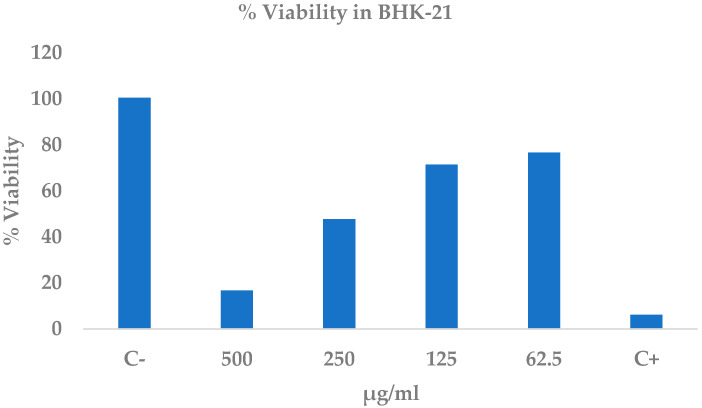
Percentage of cell viability at different concentrations of ethanolic extract of *C. procera* leaves in the BHK-21 cell line for 24 h. C+: control positive (hygromycin), C−: control negative (DMSO 0.5%).

**Figure 6 ijms-26-10574-f006:**
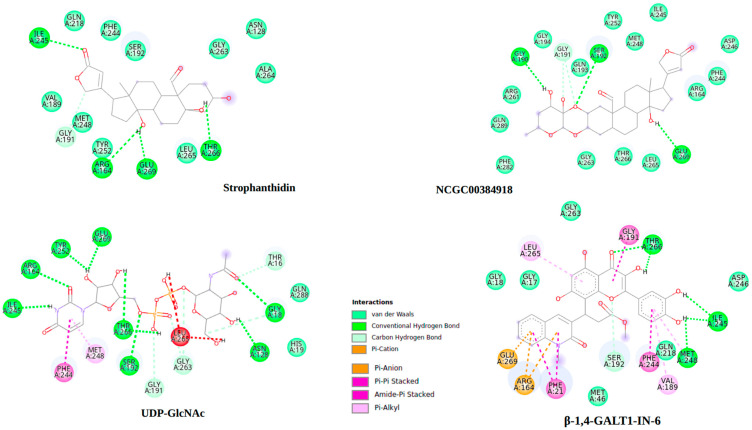
Comparison of the molecular interactions of UDP-GlcNAc, the two compounds from Calotropis extract, and β-1,4-GALT1-IN-6.

**Figure 7 ijms-26-10574-f007:**
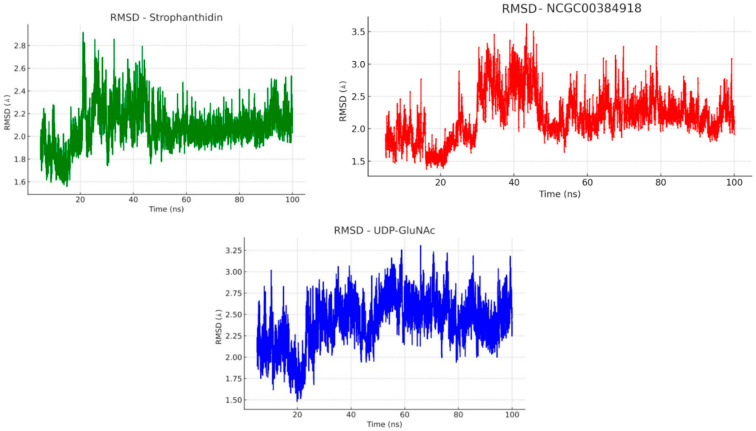
Structural stability analysis: RMSD of UDP-GlcNAc and Calotropin extract compounds in complex with MurG.

**Figure 8 ijms-26-10574-f008:**
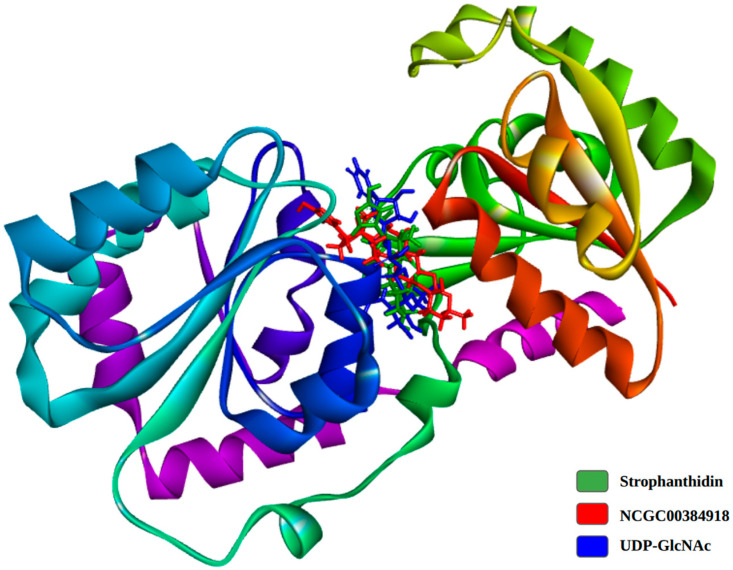
Superposition of the structures of UDP-GlcNAc and the compounds of Calotropis extract during the minimum energy state in the molecular dynamics simulation.

**Figure 9 ijms-26-10574-f009:**
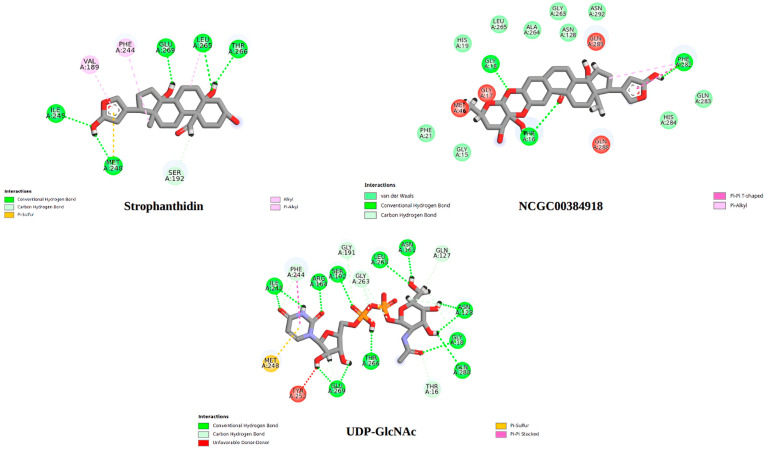
Comparison of the molecular interactions of UDP-GlcNAc, the two compounds from Calotropis extract, and β-1,4-GALT1-IN-6, during the state of minimum energy in molecular dynamics simulations.

**Figure 10 ijms-26-10574-f010:**
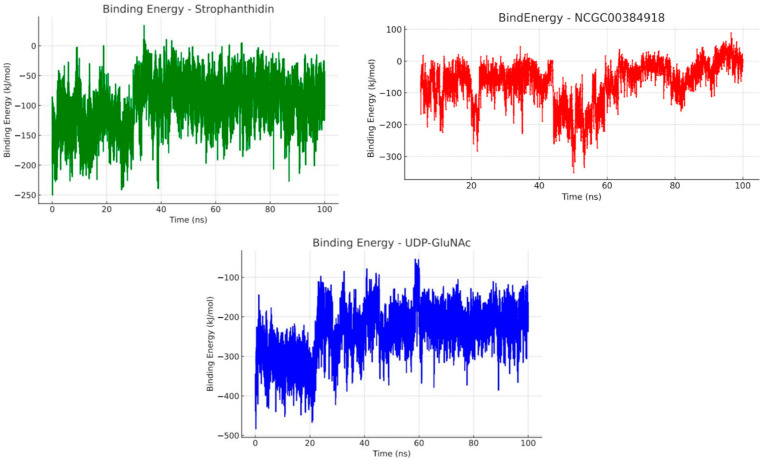
Binding energy during the dynamics of UDP-GlcNAc ligands and Calotropis extract compounds over time.

**Table 1 ijms-26-10574-t001:** Ethanolic extract of *C. procera* solubility test.

Solvent	Result
Distilled water	Insoluble
Methanol	Partially soluble
Ethanol 96%	fully soluble
Ethyl acetate	Partially soluble
Acetone	Insoluble
Chloroform	Fully soluble
Xylol	Partially soluble
Heptane	Slighty soluble
DMSO	Fully soluble

**Table 2 ijms-26-10574-t002:** Results of the phytochemical screening of the ethanolic extract of *C. procera* leaves.

Proof	Result	Methodology
alkaloids Wagner’s test	+	[16]
tannins Iron chloride (FeCl)	−	[17]
flavonoids Hydrochloric acid and magnesium tape	+	[18]
phenols Folin-Ciocalteau and sodium carbonate	+	[19]
cardiac glycosides Kiliani’s test	+	[20]
saponins Foaming	+	[21]

**Table 3 ijms-26-10574-t003:** Exact mass characteristic positive ions of compounds identified by HPLC-ESI-QTOF in *C. procera* extract.

No.	tr, min	Tentative Annotation	Formula	Observed Ion	Experimental *m*/*z*	Theoretical *m*/*z*	Δ ppm
1	10.902	Strophanthidin	C_23_H_32_O_6_	[M + H]^+^	405.2180	405.2271	−22.6169
2	11.605	Anthranilate	C_7_H_7_NO_2_	[M + H]^+^	138.0540	138.0549	−6.9175
3	11.659	Indole-3-carboxylic acid	C_9_H_7_NO_2_	[M + H]^+^	162.0530	162.0549	−12.0638
4	12.015	Indole-3-acetic acid	C_10_H_9_NO_2_	[M + H]^+^	176.0680	176.0706	−14.7952
5	12.523	NCGC00384918	C_29_H_40_O_9_	[M-H_2_O + H]^+^	515.2510	515.2639	−25.1230
6	12.892	MLS001077351-01	C_14_H_12_N_2_O_2_	[M + H]^+^	241.0930	241.0971	−17.2295
7	13.551	Tryptamine-C5:0	C_15_H_20_N_2_O	[M + H]^+^	245.1610	245.1648	−15.6629
8	14.055	Gitoxigenin	C_23_H_34_O_5_	[M + H-H_2_O]^+^	373.2300	373.2373	−19.6550
9	14.198	NCGC00179704	C_15_H_24_O_3_	[M-H_2_O + H]^+^	235.1650	235.1692	−18.0976
10	21.226	10S-Hydroxypheophorbide a	C_35_H_36_N_4_O_6_	[M + H]^+^	609.2540	609.2707	−27.5099
11	24.474	Pyropheophorbide a	C_33_H_34_N_4_O_3_	[M + H]^+^	535.2570	535.2703	−24.9724

**Table 4 ijms-26-10574-t004:** Average and standard deviation of the diameter measurement of halos in *S. aureus* and *E. coli*.

Treatments (µg/mL)	*S. aureus*	*E. coli*
1000	2.6 ± 0.7	1.6 ± 0.7
500	1.8 ± 0.7	1.5 ± 0.8
250	0.9 ± 0.1	0.8 ± 0.1
125	0.8 ± 0.1	0.7 ± 0.1
62.5	0.7 ± 0.1	0.7 ± 0.1
Positive control	2.8 ± 0.6	3.1 ± 0.3
Negative control	0.0 ± 0.0	0.0 ± 0.0

**Table 5 ijms-26-10574-t005:** Mean and standard deviation of absorbance and normalized percentage growth of *S. aureus* and *E. coli*.

Treatments (µg/mL)	*S. aureus* Absorbance (Mean ± SD)	*S. aureus* % Growth (Mean ± SD)	*E. coli* Absorbance (Mean ± SD)	*E. coli* % Growth (Mean ± SD)
1000	0.07 ± 0.02	22.61 ± 2.92	0.25 ± 0.05	35.09 ± 6.55
500	0.18 ± 0.02	40.50 ± 2.10	0.32 ± 0.01	44.49 ± 1.59
250	0.31 ± 0.04	52.71 ± 0.55	0.35 ± 0.02	48.35 ± 2.88
125	0.41 ± 0.03	59.50 ± 3.38	0.37 ± 0.00	51.93 ± 0.60
62.5	0.46 ± 0.02	61.50 ± 2.01	0.38 ± 0.01	53.61 ± 1.39
Positive control	0.12 ± 0.01	14.90 ± 1.28	0.06 ± 0.01	8.49 ± 1.49
Negative control	0.77 ± 0.01	100.00 ± 1.83	0.71 ± 0.02	100.00 ± 2.48

**Table 6 ijms-26-10574-t006:** Percentages of hemolytic activity and standard deviations according to the concentrations used.

Treatments µg/mL	% Hemolysis
Negative control	2.2 ± 0.3
1000	9.3 ± 0.1
l500	1.8 ± 1.1
250	1.9 ± 1.0
125	1.4 ± 0.6
63	0.8 ± 0.5
32	1.3 ± 0.4
Positive control	100.6 ± 1.2

**Table 7 ijms-26-10574-t007:** Percentages of cell viability and standard deviations at different concentrations of the ethanolic extract of *C. procera* leaves on human lymphocytes at 24 h. * = Statistically significant difference with respect to the negative control (PBS).

Treatments µg/mL	% Viability
500	33.2 ± 4.5 *
400	42.8 ± 9.4 *
300	52.0 ± 8.8 *
200	56.7 ± 19.5
100	63.1 ± 0.7
Methanol	41.9 ± 5.9
PBS	95.0 ± 3.8

**Table 8 ijms-26-10574-t008:** Cell viability percentages and standard deviations at different concentrations of the ethanolic extract of *C. procera* leaves on the BHK-21 cell line for 24 h. Values are shown as mean ± standard deviation. *: statistically significant difference with respect to the negative control.

Treatments µg/ mL	% Vitality
Negative control	100.5 ± 2.0
500	16.7 ± 2.3 *
250	47.7 ± 3.4 *
125	71.5 ± 8.0 *
62.5	76.7 ± 17.0 *
Positive control	6.2 ± 1.1 *

**Table 9 ijms-26-10574-t009:** The natural ligand of the MurG enzyme, UDP-GlcNAc, and the compounds belonging to the *Calotropis* extract with their respective Iso-SMILES codes and ADME-Tox level.

Compound	Iso-SMILES	Tox Level
UDP-GlcNAc	CC(=O)N[C@@H]1[C@H]([C@@H]([C@H](O[C@@H]1OP(=O)(O)OP(=O)(O)OC[C@@H]2[C@H]([C@H]([C@@H](O2)N3C=CC(=O)NC3=O)O)O)CO)O)O	6
β-1,4-GALT1-IN-6	[H]Oc1c([H])c([H])c(-c2oc3c(C@@(c4c([H])c5c([H])c([H])c([H])c([H])c5n(C([H])([H])[H])c4=O)C([H])([H])C(=O)OC([H])([H])[H])c(O[H])c([H])c(O[H])c3c(=O)c2O[H])c([H])c1O[H]	5
Alendronate	[H]OC(C([H])([H])C([H])([H])C([H])([H])N([H])[H])(P@TB1(O[H])O[H])P@SP3(O[H])O[H]	4
C6	[H]O[C@@H]1C@HOC@(C([H])([H])OP@SP3([O-])OP@SP3([O-])[O-])[C@@H]1O[H]	6
Strophanthidin	[H]OC12C([H])([H])C([H])([H])[C@]3([H])C@@(C([H])([H])C([H])([H])[C@]4(C([H])([H])[H])[C@@]3(O[H])C([H])([H])C([H])([H])[C@@]4([H])C3=C([H])C(=O)OC3([H])[H])[C@]1(C([H])=O)C([H])([H])C([H])([H])C@@(O[H])C2([H])[H]	2
Anthranilate	[H]OC(=O)c1c([H])c([H])c([H])c([H])c1N([H])[H]	4
Tryptamine-C5	[H]c1c([H])c([H])c2c(c1[H])c(C([H])([H])C([H])([H])N([H])C(=O)C([H])([H])C([H])([H])C([H])([H])C([H])([H])[H])c([H])n2[H]	3
NCGC00384918	[H]O[C@@]12O[C@@]3([H])C([H])([H])[C@@]4(C([H])=O)C@@(C([H])([H])C([H])([H])[C@@]5([H])[C@]4([H])C([H])([H])C([H])([H])[C@@]4(C([H])([H])[H])[C@]5(O[H])C([H])([H])C([H])([H])[C@]4([H])C4=C([H])C(=O)OC4([H])[H])C([H])([H])[C@@]3([H])O[C@@]1([H])OC@@(C([H])([H])[H])C([H])([H])[C@@]2([H])O[H]	2
MLS001077351-01	[H]c1c([H])c([H])c2c3c([H])c(C(=O)OC([H])([H])C([H])([H])[H])n([H])c([H])c-3nc2c1[H]	4
Indole-3-carboxylic acid	[H]OC(=O)c1c([H])n([H])c2c([H])c([H])c([H])c([H])c12	5
NCGC00179704	[H]OC(=O)C(=C([H])[H])[C@]1([H])C([H])([H])C([H])([H])[C@@]2(C([H])([H])[H])C([H])([H])C([H])([H])C([H])([H])C@(C([H])([H])[H])[C@]2([H])C1([H])[H]	5
Gitoxigenin	[H]O[C@@]1([H])C([H])([H])C([H])([H])[C@@]2(C([H])([H])[H])C@@(C([H])([H])C([H])([H])[C@@]3([H])[C@]2([H])C([H])([H])C([H])([H])[C@]2(C([H])([H])[H])C@@(C4=C([H])C(=O)OC4([H])[H])C@@(O[H])C([H])([H])[C@]32O[H])C1([H])[H]	2
Indole-3-acetic acid	[H]OC(=O)C([H])([H])c1c([H])n([H])c2c([H])c([H])c([H])c([H])c12	4

**Table 10 ijms-26-10574-t010:** Results obtained from the pharmacophoric search performed with PharmMapper.

PDB	Organism	Function	Score	PDB Code	Organism	Function	Score
NCGC00384918				Strophanthidin			
3qfw	*R. palustris*	Unknown function	0.996	1lvl	*P. putida*	Lipoamino dehydrogenase	0.929
2p67	*E. coli K12*	Transport system kinase	0.952	1nlm	*E. coli*	Essential Glucosyltransferasa	0.933
2zm5	*E. coli K12*	Transferase	0.951	1o9y	*P. syringae*	T3SS	0.921
1ghh	*E. coli*	DNA repair	0.951	2p67	*E. coli K12*	Transport system kinase	0.92
1o9y	*P. syringae*	T3SS	0.947	2zm5	*E. coli K12*	Transferase	0.916
1tu1	*P. aeruginosa*	Unknown function	0.946	1ghh	*E. coli*	DNA repair	0.916
2nya	*E. coli K12*	Nitrate reductase	0.929	1tu1	*P. aeruginosa*	Unknown function	0.909
1nlm	*E. coli*	Essential Glucosyltransferase	0.908	2nya	*E. coli K12*	Nitrate reductase	0.886

**Table 11 ijms-26-10574-t011:** Docking scores of UDP-GlcNAc, the two compounds selected from the Calotropis extract and the reference compounds.

Ligand	Binding Energy (Kcal/mol)
UDP-GlcNAc	−8.41
Strophanthidin	−9.43
NCGC00384918	−9.12
β-1,4-GALT1-IN-6	−9.29
Alendronate	−3.99
Compound 6	−6.30

**Table 12 ijms-26-10574-t012:** Average binding energy between ligands and MurG during molecular dynamics simulations.

Complex Compound-MurG	Binding Energy (kJ/mol)
UDP-GlcNAc	−237.56
Strophantidin	−96.44
NCGC00394918	−49.34

**Table 13 ijms-26-10574-t013:** Center and box sizes in Cartesian coordinates of the MurG target.

Target	X Center	X Size	Y Center	Y Size	Z Center	Z Size
MurG	38	50	−4	50	21	50

## Data Availability

Data are contained within the article.

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
