# Peer review of "Integrated In Vitro and In Silico Evaluation of the Antimicrobial and Cytotoxic Potential of Calotropis procera Leaf Ethanolic Extract: From GC-MS Profiling to Molecular Docking and Dynamics"

_ijms, 2025, doi:10.3390/ijms262110574_

Round 1

Reviewer 1 Report

Comments and Suggestions for Authors

Dear authors, I have gone through the manuscript entitled "Integrated In Vitro and In Silico Evaluation of the Antimicrobial and Cytotoxic Potential of Calotropis procera Leaf Ethanolic Extract: From GC-MS Profiling to Molecular Docking and Dynamics" and it is in the scope of the Journal. This manuscript deals with the investigation of biological activities of Calotropis procera Leaf Ethanolic Extract with emphasis on antibacterial activity.  However, through the detailed analysis of this manuscript, I noticed some omissions. Firstly, it is not clearly stated whether cytotoxicity refers to safety of use or also to anticancer activity. On the other hand in Results there is so much data that is up for discussion. Considering that the discussion is a separate section, and not together with the results, these things should be separated and the results should exclusively show the obtained results in this study. Furthermore, statistics is missing. My other comments are listed below:

  • Use Italic through all text and Figures where is necessary
  • Line 47-67 is for discussion
  • In Introduction, emphasize why plants are used as potential antimicrobial agents, why they have an advantage
  • In what were certain concentrations of extracts prepared for testing, in a medium or a solvent?
  • Is MBC determined? It would be of interest to determine MBC values of the extract
  • Line 739 – Full name of BHK-21 cells
  • In equation define components
  • 7. in English
  • Table 1 – Solubility score under table
  • Which method was used for chemical analysis since in methods and results and table is 3 different names. Decide and uniform
  • How % of inhibition in MIC was calculated? If this is already a MIC test, why are there no specific MIC values given and determined, other than % inhibition? This needs to be put into manuscript results.
  • Table 7 - what * mean? Figure 4 - there is no need to have Viability in two places. If it is average, then where are the standard deviations and statistics on the Figure? The same applies to Fig 5 and Table 8.
  • Rewrite the results, discussion, and conclusion. Avoid results in conclusion

Author Response

Dear authors, I have gone through the manuscript entitled "Integrated In Vitro and In Silico Evaluation of the Antimicrobial and Cytotoxic Potential of Calotropis procera Leaf Ethanolic Extract: From GC-MS Profiling to Molecular Docking and Dynamics" and it is in the scope of the Journal. This manuscript deals with the investigation of biological activities of Calotropis procera Leaf Ethanolic Extract with emphasis on antibacterial activity.  However, through the detailed analysis of this manuscript, I noticed some omissions. Firstly, it is not clearly stated whether cytotoxicity refers to safety of use or also to anticancer activity. On the other hand in Results there is so much data that is up for discussion. Considering that the discussion is a separate section, and not together with the results, these things should be separated and the results should exclusively show the obtained results in this study. Furthermore, statistics is missing. My other comments are listed below:

Query 1: Use Italic through all text and Figures where is necessary:

Answer:

All scientific binomial names should be italicized, e.g., Calotropis procera, Staphylococcus aureus, Escherichia coli.

Query 2: Line 47-67 is for discussion:

Answer:

Thank you for your valuable feedback. In response, we have revised the relevant section of the manuscript to improve clarity and align with the objective reporting style. Specifically, we have rephrased the paragraph to focus solely on presenting the experimental findings, including antimicrobial activity data, phytochemical composition, and relevant zones of inhibition, without incorporating interpretative or mechanistic statements.

Query 3:

In Introduction, emphasize why plants are used as potential antimicrobial agents, why they have an advantage:

Answer: We appreciate the reviewer's insightful suggestion. To address this, we have revised the Introduction to more explicitly highlight the rationale behind the use of plants as potential antimicrobial agents. We now emphasize that plants are rich sources of bioactive compounds with diverse chemical structures remarking the studied specie (C. procera). Two paragraph were added to the manuscript in order to adress the referee comment.

Query 4: In what were certain concentrations of extracts prepared for testing, in a medium or a solvent?

Answer:

Thank you for your insightful comment. The concentrations of the extracts used in our bioassays were prepared starting from a stock solution at 20,000 ppm (μg/mL). An aliquot of this stock was added into the medium to achieve the desired test concentrations. When further dilutions were necessary, additional medium was used to adjust the concentration accordingly. This approach ensured that the extracts were initially dissolved in a solvent and then appropriately diluted into the testing medium for each experiment.

We will clarify this procedure in the Methods section to eliminate any ambiguity regarding the preparation of the test solutions.

Query 5: Is MBC determined? It would be of interest to determine MBC values of the extract.

Answer:

Thank you for your valuable suggestion. In the current study, we focused on evaluating the antimicrobial activity through inhibition zones and minimum inhibitory concentration (MIC) assessments. While we did not determine the minimum bactericidal concentration (MBC) in this work, we agree that determining MBC values would provide important insights into whether the extract exhibits bacteriostatic or bactericidal effects.

We plan to include MBC testing in our future experiments and will consider adding this parameter in subsequent studies to strengthen the comprehensive evaluation of the extract's antimicrobial properties.

Query 6:

Line 739 – Full name of BHK-21 cells

Answer:

Thank you for pointing out this detail. The full name of BHK-21 cells is Baby Hamster Kidney-21 cells. We will update the manuscript at line 739 to include the complete nomenclature for clarity.

We appreciate your careful review and helpful suggestions

Query 7: In equation define components

Answer:

Thank you very much for your suggestion. Regarding, the components for the equations were defined.

Query 8:7. in English:

Answer: We want to apologize for the mistakes and typos related to the english. The article was checked again, any additional grammar-mistakes.

Query 9: Table 1 – Solubility score under table

Answer: thank you very much for your valuable suggestion. We have written the solubility score under table according your comment.

Query 10: Which method was used for chemical analysis since in methods and results and table is 3 different names. Decide and uniform:

Answer:

Query 11: How % of inhibition in MIC was calculated? If this is already a MIC test, why are there no specific MIC values given and determined, other than % inhibition? This needs to be put into manuscript results.

Answer: thank you very much for your valuable suggestion.The absorbance values obtained in the assays were converted to percentages using the negative control as the reference, since it represents the maximum bacterial growth in this medium. Although this does not alter the MIC value, it enables a clearer interpretation of the results.

Query 12: Table 7 - what * mean? Figure 4 - there is no need to have Viability in two places. If it is average, then where are the standard deviations and statistics on the Figure? The same applies to Fig 5 and Table 8.

Answer: Thank you very much for your suggestion, we added the information necessary for better comprehension.

Query 12: Rewrite the results, discussion, and conclusion. Avoid results in conclusion

Answer: Thank you very much for your suggestion. we rewrote all necessary session

Reviewer 2 Report

Comments and Suggestions for Authors

This study extracts active components of Calotropis procera using ethanol extraction methods, combining in vitro experiments with computational simulations to systematically evaluate its antibacterial effects and safety. However, the following improvements are needed:

  1. Bacteria should be italicized (S. aureus and E. coli).
  2. Lines 51-58 need to summarize the cited literature, highlighting key points.
  3. The phrase 'this study showed that ethanol' on line 64 should not discuss the experimental results in the introduction section.
  4. The result analysis should clearly describe the experimental results, while discussions about the results should be placed in the final discussion section. For example, lines 109-118 are unrelated to the results; please revise the entire document accordingly.
  5. The experimental method section has too much introductory information; please simplify it and highlight the method parts.
  6. Method 4.4, which involves collecting human blood, needs to provide ethical approval or informed consent documentation.
  7. The experimental method lacks statistical data analysis, such as Table 7 and Table 8.
  8. The figures are discontinuous, jumping directly from Figure 5 to Figure 12.

Author Response

This study extracts active components of Calotropis procera using ethanol extraction methods, combining in vitro experiments with computational simulations to systematically evaluate its antibacterial effects and safety. However, the following improvements are needed:

Query 1: Bacteria should be italicized (S. aureus and E. coli).

Answer: Thank you for your timely comments. The entire text was revised and all scientific names were corrected.

Query 2: Lines 51-58 need to summarize the cited literature, highlighting key points.

Answer: Thank you for your recommendations, they were made.

Query 3: The phrase 'this study showed that ethanol' on line 64 should not discuss the experimental results in the introduction section.

Answer: Muchas gracias por sus observaciones, estas nos permitieron mejorar la presentación de la introducción.

Query 4: The result analysis should clearly describe the experimental results, while discussions about the results should be placed in the final discussion section. For example, lines 109-118 are unrelated to the results; please revise the entire document accordingly.

Answer: Thank you for your comment, much of the results were rewritten and given the necessary direction.

Query 5: The experimental method section has too much introductory information; please simplify it and highlight the method parts.

Answer: Thank you very much for your appreciations. It was synthesized leaving what was necessary.

Query 6: Method 4.4, which involves collecting human blood, needs to provide ethical approval or informed consent documentation.

Answer: This work was under the approval of the ethics committee of the Universidad Libre Barranquilla.

Query 7: The experimental method lacks statistical data analysis, such as Table 7 and Table 8.

Answer: Statistical analysis methods were carried out to determine the statistically significant differences between the different treatments. These were discussed in the text.

Query 8: The figures are discontinuous, jumping directly from Figure 5 to Figure 12.

Answer: Thank you for your contribution, many apologies for the error, we corrected everything necessary.

Reviewer 3 Report

Comments and Suggestions for Authors

This study explores the antibacterial effects and safety of an ethanol based leaf extract from Calotropis procera. By combining lab experiments and MD and docking methods, the authors show that the extract can effectively fight Staphylococcus aureus and Escherichia coli. This is an interesting topic, and the manscript is generally well-organized, but needs clarifications. The result part consists heavily of analytical methods.  The organization of the result and discution sections is too fragmented. I think this sections need improvements. Some minor comments:

Figure 13 shows the RMSD plots in nm. This must be Angstrom. Additionally the authors claim UDP-GlcNAc, Strophanthidin, and NCGC00384918 reached a stable equilibrium after 25ns. This seems to be after 50 ns. 
Average binding energy between ligands and MurG during molecular dynamics simulation in table 15 is shown with positive sign.
Figure 16 shows the binding energy in kj/mol, but the table 15 shows the value kcal/mol. Something is wrong here. 

I hope this is helpful. 

Author Response

This study explores the antibacterial effects and safety of an ethanol based leaf extract from Calotropis procera. By combining lab experiments and MD and docking methods, the authors show that the extract can effectively fight Staphylococcus aureus and Escherichia coli. This is an interesting topic, and the manscript is generally well-organized, but needs clarifications. The result part consists heavily of analytical methods.  The organization of the result and discution sections is too fragmented. I think this sections need improvements. Some minor comments:

Query:

Figure 13 shows the RMSD plots in nm. This must be Angstrom.

Answer:

Thank you very much for your valuable comment. We appreciate your careful review and helpful suggestions. As you correctly pointed out, there was an inconsistency in the units used in Figure 13. While we conducted the measurements and RMSD analysis in angstroms (Å), the figure mistakenly displayed the units as nanometers (nm). We have corrected this error, and Figure 13 now properly reflects the units in angstroms. We apologize for the oversight and thank you again for bringing it to our attention.

Query:

Additionally the authors claim UDP-GlcNAc, Strophanthidin, and NCGC00384918 reached a stable equilibrium after 25ns. This seems to be after 50 ns.

Answer:

Thank you very much for your insightful comment. We appreciate your observation regarding the equilibrium point in our simulation. Initially, we stated that equilibrium was reached around 25 ns. However, as you correctly pointed out, not all molecules may have achieved energetic equilibrium at that time. Taking this into account, we have revised line 545 to reflect a more accurate estimation, establishing the equilibrium point at 50 ns. We are grateful for your suggestion, which helped us improve the clarity and accuracy of our analysis.

Query:

Average binding energy between ligands and MurG during molecular dynamics simulation in table 15 is shown with positive sign.

Answer:

Thank you very much for your valuable comment. You correctly pointed out an important inconsistency in the presentation of the average binding energy values. In Table 15, the binding energy values were mistakenly shown with a positive sign. This was an error, as the results obtained prior to the MD simulation indicated that these values should be negative. We have corrected this mistake by updating the values in Table 15 to reflect the correct negative sign. Additionally, we revised paragraph 615–621 and line 641, where the binding energy was again incorrectly presented as positive. All relevant values are now consistently shown with the correct negative sign. We sincerely apologize for the oversight and thank you for helping us improve the accuracy of our manuscript.

Query:

Figure 16 shows the binding energy in kj/mol, but the table 15 shows the value kcal/mol. Something is wrong here.

Answer:

We would like to sincerely apologize for the oversight regarding the units of energy in our manuscript. The energy values obtained from the MD simulation should have been reported in kJ/mol. However, in Table 15, these values were erroneously presented in kcal/mol. To improve the consistency and accuracy of the paper, we have corrected Table 15 and now present the average binding energy values in kJ/mol. Additionally, we have revised lines 615–621, line 641, and Figure 16, where the energy units were also incorrectly shown as kcal/mol. All references to binding energy of the MD simulation throughout the manuscript are now consistently expressed in kJ/mol. We appreciate your attention to detail and thank you for helping us enhance the quality of our work.

Round 2

Reviewer 1 Report

Comments and Suggestions for Authors

I recommend manuscript for publication. 

Author Response

We appreciate your feedback; it has significantly improved the presentation of our manuscript.

Reviewer 2 Report

Comments and Suggestions for Authors

The author's manuscript has been revised to meet the requirements. Please add an ethics statement or informed consent in the manuscript.

Author Response

We appreciate your comments. Informed consent was obtained from all participants. In addition, the volunteers provided written consent for the publication of this article. This study did not involve the direct participation of patients or the collection of clinical information. Biological samples (human blood for hemolysis assays) were obtained from healthy adult volunteers who provided informed consent, in accordance with national regulations (Resolution 8430 of 1993, Colombia) and the guidelines of the Ethics Committee of Universidad Libre.